# Developmental Biology and Seasonal Damage of the Grape Borer *Xylotrechus pyrrhoderus* in Grapevines

**DOI:** 10.3390/insects16090979

**Published:** 2025-09-19

**Authors:** Ganyu Zhang, Yuying Jia, Haibin Wu, Yong Zhang, Murad Ghanim, Yanan Ma, Ruihong Sun

**Affiliations:** 1Shandong Institute of Pomology, Shandong Academy of Agricultural Sciences, Tai’an 271000, China; zganyu@126.com (G.Z.); jinghaijiangxuan@126.com (H.W.); gkszhyong@sina.com (Y.Z.); 2College of Plant Protection, Shandong Agricultural University (Main Campus), Tai’an 271000, China; jiayy0321@163.com; 3Department of Entomology, Volcani Institute, Rishon LeZin 7505101, Israel; ghanim@volcani.agri.gov.il

**Keywords:** *Xylotrechus pyrrhoderus*, vineyards, X-ray imaging, micro-computed tomography

## Abstract

The grape borer *Xylotrechus pyrrhoderus* is a major pest in viticulture. Its larvae tunnel into grapevine canes, damaging internal tissues, disrupting nutrient transport and causing cane withering, reduced yield, and poor fruit quality. However, limited knowledge of its biology has made it difficult to develop effective control strategies. To address this gap, field surveys were conducted and imaging tools such as X-ray and micro-CT were used to study the pest’s damage patterns and life cycle. The findings revealed that infestation rates were significantly higher in May compared with December, highlighting the importance of combining spring pruning with traditional winter practices to better manage the pest. Internal damage to grape canes and morphological features of the grape borer across all life stages (larva, pupa, and adult) were clearly visualized and the morphological features of the insect at different stages were documented. Notably, adults remained inside the canes for about two weeks after emergence, and were smaller in body size, possibly as an adaptation to narrow pupal chambers. These findings enhance our understanding of *X. pyrrhoderus* biology and offer practical insights for integrated pest management and developing effective control strategies. Ultimately, this research supports more sustainable and environmentally friendly grape production practices.

## 1. Introduction

According to the Food and Agriculture Organization, China is a major grape-growing and wine-producing country, with extensive vineyards and high production. In contrast, grape cultivation and wine industries in Japan and Korea are relatively small. The grape borer *Xylotrechus pyrrhoderus* Bates (*Coleoptera: Cerambycidae*) is a major and significant pest in grapevine cultivation. When the insect colonization is high, the infestation rate of grape plants can reach over 85% [1,2,3,4,5]. *X. pyrrhoderus* is distributed across Asia, with confirmed occurrence in China, Korea, and Japan. In China, it is prevalent in numerous regions, including Jilin, Liaoning, Shanxi, Shaanxi, Shandong, Hebei, Henan, Zhejiang, Jiangsu, Jiangxi, Anhui, Sichuan, Guizhou, Fujian, Guangxi, and various other provinces, municipalities, and autonomous regions [4,5,6,7,8,9,10,11,12,13]. *X. pyrrhoderus* completes one generation per year and primarily targets grape canes. Adults are diurnal and most active between 10:00 and 14:00 h during August to September [14]. On the day of mating, the female can lay eggs in the crevices of grape bud scales or on grape canes. Each female lays one egg at a time. The egg stage lasts approximately 5–7 days, and each female lays between 30 and 90 eggs [12]. The pest overwinters as young larvae in damaged grape canes. Larval development resumes in April and May, during which larvae feed internally. This feeding can prevent grape buds from bursting or cause them to dry after bursting, and in severe cases, lead to dieback of multiple canes and vines. The larval tunnels are typically filled with frass and sawdust-like debris, further contributing to cane decline [12,15]. Pupation occurs from early July to the end of August, and newly emerged adults remain in the canes for a period before leaving. Mating may occur on the same day the adults emerge. Sex pheromones, specifically (2S,3S)-octanediol (**1**) and (2S)-hydroxy-3-octanone, have been identified as key components in the mating behavior of play an important role in the mating behavior of *X. pyrrhoderus* [1,14,16].

*X. pyrrhoderus* was first documented in 1954 during a comparative study of the genitalia of adult male longhorn beetles [17]. Since the 1980s, research on this species has gradually increased [1,14,16,18,19,20]. However, much of this research remains limited in scope and in somewhat outdated, offering only a partial understanding of the species’ biology and providing insufficient guidance for effective prevention and control. In recent years, infestations of *X. pyrrhoderus* have become increasingly widespread, particularly in China, hindering the sprouting of new canes, causing extensive vine dieback, and severely affecting grape production and quality. In heavily damaged vineyards, the damage rate can exceed 85% largely due to the lack of effective control measures [12]. Although no effective control methods are currently available for *X. pyrrhoderus*, the identification of sex pheromones in this and related species—where they have been successfully used for monitoring and control—along with the previous application of microbiological agents such as entomopathogenic fungi against closely related taxa, suggests that these approaches warrant further investigation and potential adoption for managing *X. pyrrhoderus*. Many life stages of boring pests (eggs, larvae, and pupae) occur within the branches and trunks, making direct observation difficult. Researchers have relied on dissecting host material to study insect development; however, this approach disrupts the insects’ natural environment and may influence its growth and behavior. In recent years, X-ray imaging and micro-computed tomography (micro-CT) have gained popularity in agriculture and forestry research for nondestructive evaluation tools. These techniques enable early detection of internal structural changes and provide detailed insights into insect morphology and development without disturbing their habitat. X-ray imaging has been successfully used to detect split pits in peaches [21], water core and browning in apples [22,23], and various physiological disorders caused by insect or fungal infestations [24,25,26,27,28,29,30,31]. Similarly, micro-CT has been used extensively, helping to advance morphological and physiological studies, including those investing insect anatomy and morphology [32], brain structure and development [33,34], real-time growth [35], and reproductive systems [36].

The biological characteristics of *X. pyrrhoderus* remain poorly understood, and effective control strategies are currently lacking. A major challenge, except for the adult, develop concealed within grapevine canes, making detection and study difficult. Therefore, advancing the current understanding of *X. pyrrhoderus* biology is essential for the development of effective control strategies. The present study used X-ray and micro-CT, two widely used non-destructive technologies, to observe and analyze the damage characteristics and development of *X. pyrrhoderus* in grape canes. These methods offer valuable insights into the pest’s hidden life stages without disrupting the natural environment, thus supporting more informed pest management approaches.

## 2. Materials and Methods

### 2.1. Study Area

The study was conducted at the vineyard of Wanji Mountain Base, Shandong Institute of Pomology (36°12′ N, 117°4′ E), Tai’an City, Shandong Province, China. The vineyard was isolated separately for the experiment without spraying any pesticides.

### 2.2. Investigation of Damaged Canes

Grape canes were surveyed from December 2022 to May 2025. During this period, the number of damaged canes and plants was recorded, and the damage characteristics were described. Beginning December 2023, the vineyard was divided into three treatment groups (untreated Control, winter pruning, winter + spring pruning). Each group consisted of 10 rows with approximately 30 vines per row. The extent of damage to grapevines under different treatment conditions was investigated. The traits of cane diameter and vigor were not assessed in this study.

### 2.3. Micro-CT of Damaged and Healthy Canes

15 damaged and 15 healthy grape canes, each approximately 20 cm long, were collected in the field. The positions of larvae in damaged canes were preliminarily identified using X-ray imaging. Three damaged and three healthy grape canes were selected for the following experiments. Although the number of tested samples was small (three damaged and three healthy), the quality of the results indicates that the selected samples were representative of the observed damage, allowing the findings to be generalized to other damaged samples in the field. Samples approximately 4 cm in length were cut from damaged canes (without larvae) and healthy canes for analysis. The samples were scanned using an X-ray microscope (nanoVoxel-3000, Sanying Precision Instrument Co., Ltd., Tianjin, China). Each sample was affixed to a carbon fiber rod, clamped with a gripper, and adjusted to ensure that the measurement area remained within the field of view throughout the test. After allowing to stand for 10 min, the CT (nanoVoxel-3000, Sanying Precision Instrument Co., Ltd., Tianjin, China) test was initiated. Test conditions included 60.0 kV, 80.0 μA, and 0.60 s exposure duration. The scanning data were transmitted to a back-end computer and imported into Avizo 9.0.1 software (Thermo Fisher Scientific, Waltham, MA USA). Avizo Software 3D Visualization and Analysis. Available online: https://www.thermofisher.cn/cn/zh/home/electron-microscopy/products/software-em-3d-vis/avizo-software.html (accessed on 18 March 2023). for quantitative analysis and comparison.

Similarly, damaged canes containing larvae were cut into 4 cm sections and scanned under identical conditions. Data were processed using Avizo 9.0.1 for further analysis and comparison.

### 2.4. Tracking and Photographing the Development Status of X. pyrrhoderus in Canes Using X-Ray Imaging

Grape canes containing larvae were collected from the field and dissected in the laboratory. Larvae were then transferred to healthy and fresh canes to continue feeding. The developmental characteristics of the grape borers at different stages were tracked and observed daily using photographs captured via X-ray imaging. In this experiment, grape borers in 160 canes were tracked and observed. For imaging, canes were vertically fixed in test tubes and mounted on the rotating platform of a 3D X-ray microscope (nanoVoxel-2000, Sanying Precision Instrument Co., Ltd., Tianjin, China). The rotating center of the sample was adjusted to locate the insect’s position. Imaging parameters were set to 130.0 kV voltage, 400.0 μA current, and 0.10 s exposure duration. X-ray photographs were used to describe the characteristics of different developmental stages of *X. pyrrhoderus*. The sizes of the insects at different stages and dimensions of their tunnels were measured and compared.

### 2.5. Data Analysis

Data analysis was performed using IBM SPSS 25.0 (IBM Corp, Armonk, New York, USA). Independent sample *t*-tests were used to compare grape canes and plant damage between December and May as well as the porosity of healthy versus damaged canes. One-way analysis of variance (ANOVA) was used to compare body length, body width, and developmental time of different insect stages within canes. Pearson correlation analysis, performed in GraphPad Prism 9 (GraphPad Software, Inc., La Jolla, CA, USA), was used to examine the relationship between insect body size and pupal compartment size.

For all parametric tests, assumptions were evaluated prior to analysis. Normality was assessed using the Shapiro–Wilk test and inspection of residual plots, and homogeneity of variances was tested with Levene’s test. When assumptions were violated, data were log-transformed; if transformation did not resolve the issue, non-parametric alternatives were considered. Graphs were created using GraphPad Prism 9, and image tagging and processing were performed using Photoshop 2021 (Adobe Systems, Inc., San Jose, CA, USA).

## 3. Results

### 3.1. External Characteristics of Damaged Canes

*X. pyrrhoderus* primarily damages grapevine canes through its larvae. The initial signs of damage are subtle and easily overlooked. As larvae grow and start feeding, the bark at damaged sites turns black during winter (Figure 1A,B), representing the first visible symptom of damage. From April to May of the following year, young larvae become increasingly active and feed on the xylem and pith tissues. This feeding disrupts vascular function, leading to cane wilting or dieback. Larval tunnels are typically packed with frass and sawdust. Damaged grape cane buds either fail to germinate or die shortly after bursting, resulting in severe cane dieback, which is the second visible indicator of infestation (Figure 1C,D). Damage surveys conducted in December 2022 and May 2023 showed that the rate of damaged canes and plants was significantly higher in May (F = 6.53, df = 15.75, *p* < 0.05; F = 0.14, df = 28, *p* < 0.05; Figure 2A). In Figure 2B, the assessment of grapevine damage under different treatment conditions showed that the damage rate of canes increased sharply without any treatment (control) against *X. pyrrhoderus*. Specifically, as *X. pyrrhoderus* developed and continued to feed, and this trend in the damage rate became even more pronounced. However, the combination of winter and spring pruning (w + s p), led to a reduction in grapevine damage, indicating that this integrated approach is more effective in mitigating the impact of *X. pyrrhoderus*.

### 3.2. Internal Characteristics of Damaged Canes

The larvae of *X. pyrrhoderus* feed beneath the bark, initially damaging the phloem and subsequently burrowing into the xylem, forming irregular curved tunnels (Figure 3). Although external symptoms such as blackened bark becomes less apparent as canes mature, micro-CT imaging revealed clear evidence of internal structural damage. The affected areas displayed distinct signs of larval feeding activity. Compared with healthy canes (Figure 4A,C,E), infested canes exhibited severe disruption to phloem and xylem vascular tissues (Figure 4B,D,F), rendering these tissues nonfunctional. Quantitative analysis further supported these experiments: the porosity of healthy canes was significantly higher (30.70% ± 0.36%) than that of damaged canes (27.63% ± 0.31%; F = 0.154, df = 4, *p* = 0.003). Notably, the calculated porosity of the damaged canes included the larval tunnels, yet, the porosity of these canes was significantly lower than that of healthy ones.

### 3.3. Developmental Stages and Morphological Characteristics of X. pyrrhoderus Within Grapevine Canes as Revealed by X-Ray Image Analysis

To investigate the developmental morphology of *X. pyrrhoderus*, damaged canes were monitored using X-ray imaging. The results revealed the presence of a distinct prepupal stage between the larval and pupal stages (Figure 5B). The observations at different stages are summarized as follows: (1) Larva: characterized by a slender, cylindrical body with a large head, well-adapted for movement and feeding within canes (Figure 5A). (2) The prepupa has a cylindrical body with distinct segmentation, is immobile and non-feeding (Figure 5B). (3) The pupa is visible and located within pupal chamber, with clearly defined head, thorax, and abdomen. The abdominal end exhibit some movement, mainly swinging back and forth (Figure 5C). (4) Adult, fully formed with segmented head, thorax, and abdomen, along with visible legs and high mobility (Figure 5D). Developmental timing analysis showed that the adult had the longest developmental duration (14.00 ± 0.34 days) aside from the larval period, which was significantly longer than that of the prepupal (8.07 ± 0.25 days) and pupal (9.87 ± 0.29 days) stages (F = 106.66, df = 2, *p* < 0.05; Table 1).

Using X-ray imaging and Avizo 9.0.1 software, the body lengths and widths of *X. pyrrhoderus* prepupae, pupae, and adults were measured. Prepupae were the longest (12.44 ± 0.26 mm), followed by pupae (12.14 ± 0.33 mm) and adults (10.36 ± 0.29 mm) (Table 1). Adult body length was significantly shorter than that of the prepupae and pupae (F = 13.987, *p* < 0.05), with no significant difference between prepupal and pupal lengths. Prepupal, pupal, and adult body widths were 2.88 ± 0.11, 2.81 ± 0.09, and 2.43 ± 0.06 mm, respectively. Adult body width was significantly smaller than prepupal and pupal body widths (F = 7.455, *p* = 0.001), with no significant difference between prepupae and pupae.

Correlations between body dimensions and pupal chamber size were also analyzed to explore potential spatial constraints during development. Significant positive correlations were found between prepupal, pupal and adult body length, and the length of the pupal chamber (Pearson r = 0.8075, 0.8371, 0.8881, respectively, *p* < 0.001, Figure 6A–C). Similarly, both pupal and adult body width were significantly correlated with the pupal chamber width (Pearson r = 0.7906, 0.7479, *p* < 0.001, Figure 6D–F). Among these, adult body length showed the strongest correlation with pupal chamber length, suggesting that adult body size may be closely influenced by the spatial constraints of the chamber. In contrast, prepupal body width was not significantly correlated with the width of the pupal chamber, indicating variability in body width during the prepupal stage may not be limited by chamber dimensions (Pearson r = 0.3737, *p* > 0.05).

## 4. Discussion

In this study, we investigated the damage, symptoms and developmental characteristics of *X. pyrrhoderus* in grapevine canes using field surveys, X-ray imaging, and micro-CT analysis. Field observations confirmed that the presence of larvae can often be identified by blackened bark of grape canes indicative of larval feeding activity. However, blackened bark of grape canes has limitations. Not all infested canes exhibit visible signs early in infestation, and bark blackening can also result from unrelated factors such as frost damage or fungal disease, thus reducing diagnostic reliability [37,38]. To improve early detection and control, our results support combining winter pruning of visibly damaged canes [36], with additional removal of wilting canes during spring and summer. The larvae have limited mobility and remain within the original canes. By the following spring, they continue feeding on these same canes, causing further damage. The purpose of spring pruning is that the injury caused by larvae during the previous winter is often not immediately visible. Some affected areas may go undetected during winter pruning. However, by the next spring, the damage becomes more apparent, allowing infested canes to be identified, pruned, and removed. Timely disposal of these canes is crucial as it can significantly reduce larvae and prevent further infestation and damage [3,15].

The use of X-ray imaging and micro-CT in this study not only visualized the extent of internal damage caused by *X. pyrrhoderus* but also revealed that larval feeding directly disrupts vascular tissues, leading to functional decline and cane dieback. These imaging techniques allowed non-destructive observation of developmental stages, enabling documentation of the morphology and duration of prepupal, pupal, and adult phases. This pattern of injury inside canes contrasts with many other wood-boring pests, where damage is typically restricted to gallery formation in the heartwood. Examples include visualizing the gallery systems of drywood termite species like *Cryptotermes secundus* [39] and *Incisitermes minor* [40], as well as the development of the oak platypodid beetle *Platypus quercivorus* in wood [41]. In addition, micro-CT technology has also been applied to the study of the internal morphology of some insects, such as the analysis and comparison of the internal structure of the rostra of weevils [42], and the morphological characteristics of the larvae, pupae and adults of *Chrysopa pallens* [43]. Such differences likely reflect species-specific feeding behaviors, life cycle strategies, and host–plant interactions, underscoring the unique ecological impact of *X. pyrrhoderus*. Importantly, the application of these imaging techniques enabled us to document developmental stages non-destructively, providing new opportunities for linking morphological traits with life-history timing. Unlike previous concerns in the literature [44,45], our results further indicate that the levels of X-ray irradiation used during scanning did not interfere with development, although subtle physiological effects still merit investigation. Overall, these findings highlight both the distinctive biological impacts of *X. pyrrhoderus* and the value of imaging technologies for advancing our understanding of its pest status.

Boring pests, particularly cerambycid beetles, share several common traits, including long life cycles, irregular adult emergence, and concealed larval habits. Their life cycle typically consists of four distinct stages: egg, larva, pupa, and adult. However, considerable variation exists among cerambycid species. For instance, *Monochamus alternatus* has a pupal stage lasting 11–24 days, an egg stage of 5–12 days, and a larval period of approximately 216.6 days [46]. In *Glenea cantor*, the egg stage lasts around 5.13 days, the larval stage about 41.13 days, the pupal stage 10.96 days, and newly emerged adults remain inside branches for 5.96 days [47]. Similarly, in *Anoplophora glabripennis*, the pupal stage lasts 15.05 days, the prepupal stage 5.05 days, the larval stage 153.78 days, and adults stay within branches for about 7 days after emergence [48].

The observed developmental timeline of *X. pyrrhoderus* reveals traits affecting its pest potential. Extended periods within canes during pupation and after adult emergence indicate prolonged resource use and a longer period during which host plants are vulnerable. By documenting these dynamics under natural, non-destructive conditions, our study provides a more realistic view of the species’ life history than earlier research relying on artificial rearing or destructive sampling. This approach not only refines our understanding of *X. pyrrhoderus* biology but also underscores broader implications for cerambycid ecology and taxonomy, where life-cycle variation can reflect key adaptations to host–plant interactions. The ability to monitor development without disrupting the insect or its substrate opens new avenues for studying boring pests in ecologically relevant contexts. This study revealed a notable behavior of *X. pyrrhoderus* adults as they remained inside the grape canes for approximately 14 days after eclosion, consistent with earlier observations [19]. Considering the grapevine’s growth cycle, this occurs during the period when the grapes are starting to change color. These 14 days residence of *X. pyrrhoderus* adults within the canes is the longest compared to the other cerambycid studied so far. This finding highlights a critical window for targeted pest control, before adults emerge and begin reproduction. X-ray imaging showed that the prepupal stages were immobile, pupae exhibited limited movement, and adults highly mobile within canes after eclosion. Morphologically, adults were significantly smaller in both length and width compared to pupae and prepupae (Table 1), likely reflecting adaptation to spatial constraints within the pupal chambers. This was further supported by strong correlation between adult body size and chamber dimensions. Interestingly, in this study we found *X. pyrrhoderus* adults were capable of mating and ovipositing on the same day they exited the canes, following the 14-day intra-cane period. Premating time in insects is often correlated with sexual maturity and physiological development [49,50]. Previous research has indicated that for *Glenea cantor*, the time spent inside branches is related to its sexual maturation [47]. Future studies should investigate whether the 14-day intra-cane period supports sexual maturation and reproductive readiness in *X. pyrrhoderus*, as well as serving a protective function.

While this study successfully tracked the prepupal, pupal, and adults duration of development, larval development remains poorly documented due to its extended duration spanning over 300 days from the hatching in August to maturation in June or July of the following year [51]. Additionally, data on larval instars are currently lacking in this and other studies, representing a limitation for both the present work and its overall conclusions. Current laboratory rearing techniques are only capable of supporting mature larvae through to adulthood, as the survival rate of early instar larvae under indoor conditions is low. These early stages appear to require environmental conditions unique to vineyards, such as host cane structure and microclimate. Therefore, larvae need to be collected from the field at later developmental stages when they reach maturity the following year. However, by the time of collection, substantial damage to grapevines has already occurred. Thus, at present, conducting research on *X. pyrrhoderus* is highly time-consuming and expensive. To advance our understanding and improve management of this pest, the development of reliable indoor rearing methods is essential. Successful artificial rearing would facilitate year-round studies of all life stages and allow for controlled experimentation on behavior, physiology, and control strategies [52,53]. Currently, information on the larval biology of *X. pyrrhoderus* is lacking, highlighting a knowledge gap that warrants further investigation. Our current efforts are focused on improving the artificial rearing techniques for *X. pyrrhoderus*, ensuring the survival of all life stages. These advancements will facilitate deeper exploration of this pest’s biology and behavior, contributing to more effective control measures and improved grape yield and quality.

## 5. Conclusions

This study investigated the biological characteristics of *X. pyrrhoderus* by comparing grapevine cane damage. The findings highlight several points of practical importance for vineyard management: (1) Damage peaks in spring/summer, not just winter: Cane damage was significantly higher in May, underscoring the need to extend pest management beyond winter pruning to include control measures during the growing season. (2) Hidden development stage creates a control window: Adults remain inside canes for about two weeks before emergence, providing an opportunity for targeted interventions during the grape enlargement and color-change stages. (3) Cane structure and insect size are interconnected: Damaged canes had reduced porosity, and adult size was closely tied to pupal chamber length, linking internal cane damage to insect development and offering insights for monitoring. (4) Given that *X. pyrrhoderus* is restricted to Asia, it is important to develop monitoring programs outside the region to enable precautionary measures if the species is detected on other continents.

## Figures and Tables

**Figure 1 insects-16-00979-f001:**
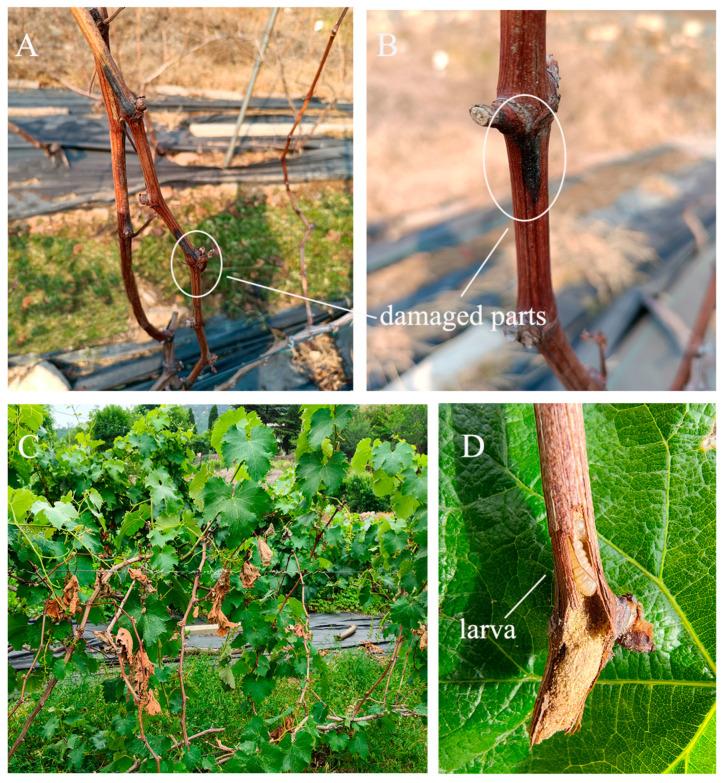
External characteristics of grape canes and plants damaged by *X. pyrrhoderus*. (**A**) Overall morphological characteristics of grape canes and plants damaged by *X. pyrrhoderus* in winter; (**B**) Local morphological characteristics of grape canes damaged by *X. pyrrhoderus* in winter; (**C**) Overall morphological characteristics of grape canes and plants damaged by *X. pyrrhoderus* in in spring; (**D**) Local morphological characteristics of grape canes damaged by *X. pyrrhoderus* in winter.

**Figure 2 insects-16-00979-f002:**
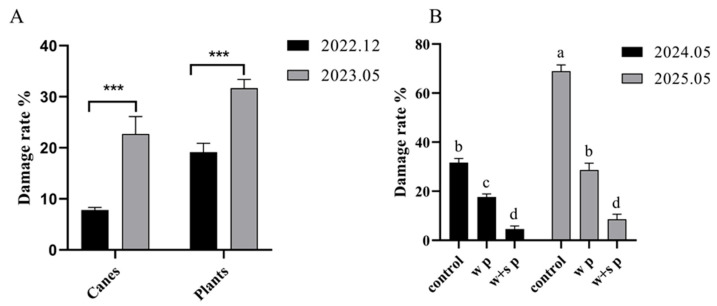
Damage caused by *X. pyrrhoderus* on grape canes and plants. (**A**), Damage rate of grape canes and plants (mean ± SE; *** *p* < 0.05). (**B**), Damage rate of grape plants under different treatments (mean ± SE; different letters “a,” “b,” “c”, “d” indicate statistically significant differences).

**Figure 3 insects-16-00979-f003:**
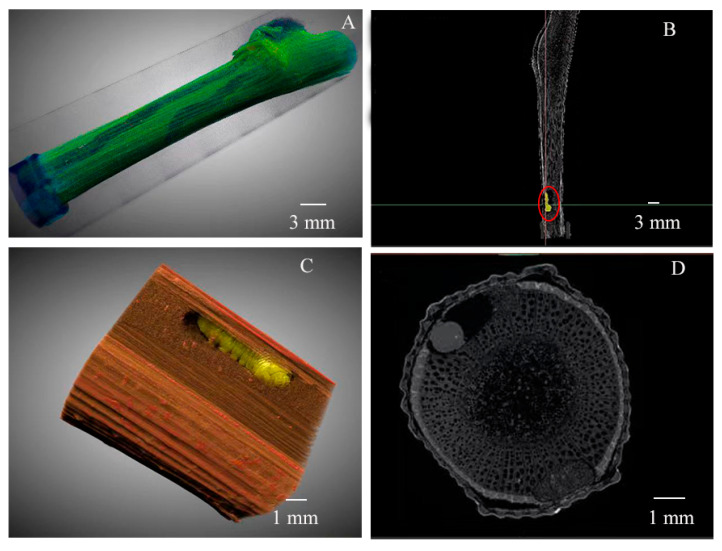
Internal characteristics of grape canes damaged by *X. pyrrhoderus*, showing larvae and feeding tunnels within the tissue. (**A**), Three-dimensional rendering of infested canes. (**B**), Longitudinal CT section of an infested cane, the red circle indicates the three-dimensional reconstruction of the larvae of *X. pyrrhoderus*. (**C**), Three-dimensional reconstruction of an infested cane. (**D**), Transverse CT section of an infested cane (*n* = 12).

**Figure 4 insects-16-00979-f004:**
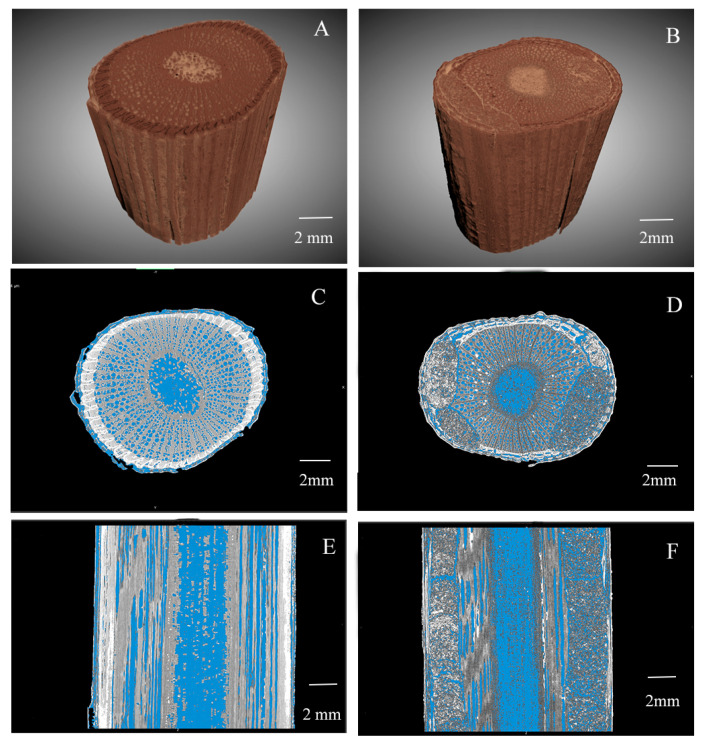
Internal characteristics of grape canes. (**A**,**C**,**E**) Healthy canes. (**B**,**D**,**F**) Canes damaged by *X. pyrrhoderus* (*n* = 6). The internal pores of the canes are visualized in blue.

**Figure 5 insects-16-00979-f005:**
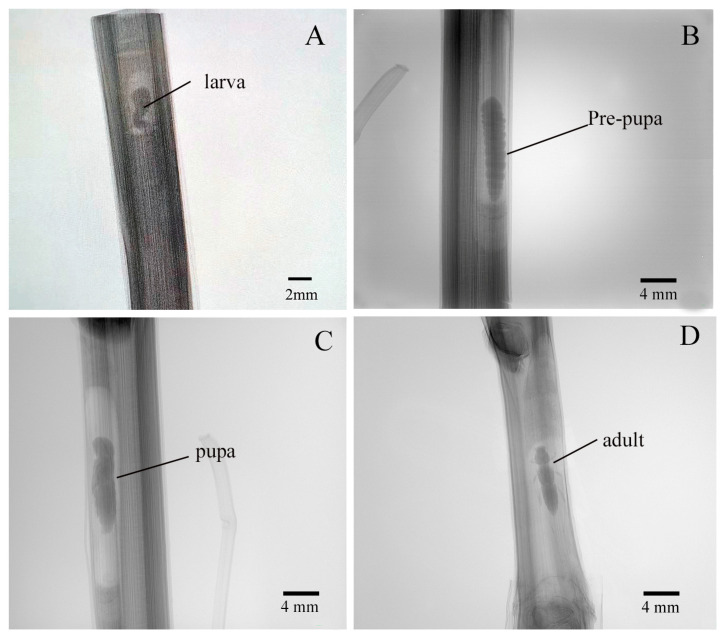
Developmental stages and morphological characteristics of *X. pyrrhoderus* in grape canes. (**A**), Larval stage. (**B**), Prepupal stage. (**C**), Pupal stage. (**D**), Adult stage. *n* indicates the number of samples analyzed (*n* = 160).

**Figure 6 insects-16-00979-f006:**
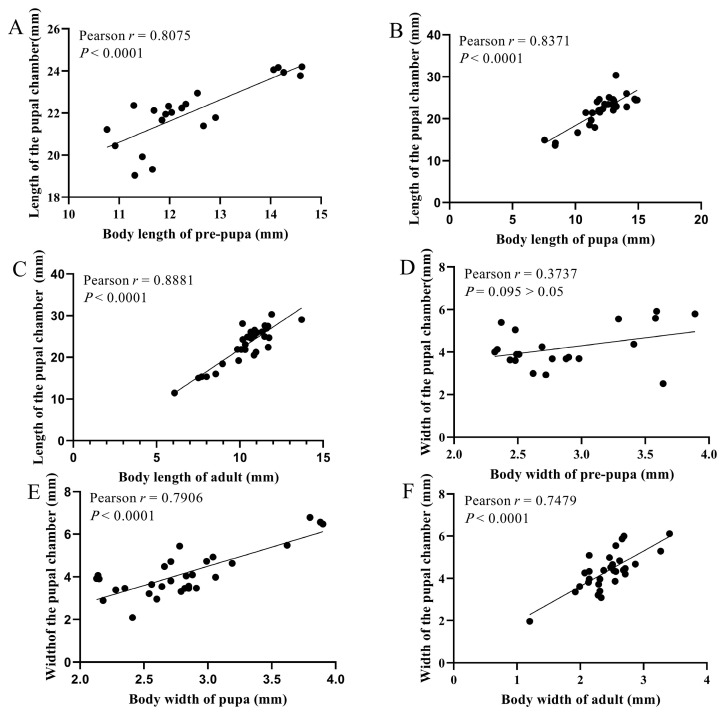
Correlation between body size and pupal chamber size of *X. pyrrhoderus* at different developmental stages. (**A**), Pre-pupal length vs. pupal chamber length. (**B**), Pupal length vs. pupal chamber length. (**C**), Adult length vs. pupal chamber length. (**D**), Pre-pupal width vs. pupal chamber width. (**E**), Pupal width vs. pupal chamber width. (**F**), Adult width vs. pupal chamber width.

**Table 1 insects-16-00979-t001:** Comparison of the size and development time of *X. pyrrhoderus* at different developmental stages.

Developmental Stage	Body Length (mm)	Body Width (mm)	Time Spent Within Canes (Days)	Activity Signs
prepupa	12.44 ± 0.26 ^a^	2.88 ± 0.11 ^a^	8.07 ± 0.25 ^c^	immobile
pupa	12.14 ± 0.33 ^a^	2.81 ± 0.09 ^a^	9.87 ± 0.29 ^b^	the abdominal end exhibits some movement
adult	10.36 ± 0.29 ^b^	2.43 ± 0.06 ^b^	14.00 ± 0.34 ^a^	strong mobility

Mean ± SE, *p* < 0.05, different letters “a” “b” “c” indicate statistically significant differences.

## Data Availability

The data presented in this study are available on request from the corresponding authors.

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
