# Peer review of "Developmental Biology and Seasonal Damage of the Grape Borer *Xylotrechus pyrrhoderus* in Grapevines"

_insects, 2025, doi:10.3390/insects16090979_

Round 1

Reviewer 1 Report

Comments and Suggestions for Authors

This manuscript explores the developmental biology and damage characteristics of the grape borer Xylotrechus pyrrhoderus through a combination of field surveys, X-ray imaging, and micro-computed tomography (micro-CT). It offers a comprehensive analysis of both internal and external damage to grapevines, while providing new insights into the pest’s life cycle and morphology. The study is timely, methodologically innovative, and contributes valuable knowledge to the field of sustainable pest management in viticulture.

In my view, the manuscript is scientifically sound and presents meaningful findings, but it requires some revisions before it can be accepted for publication. Specifically, the manuscript would benefit from language refinement and improved grammatical consistency. While the overall structure is acceptable, reorganizing certain sections could further enhance clarity, though this is not essential. Some aspects of the methodology also require additional clarification to ensure transparency and reproducibility.

The manuscript’s major strengths lie in its novelty and relevance. It addresses a clear knowledge gap concerning the biology and management of X. pyrrhoderus, and it makes excellent use of underutilized, non-destructive imaging technologies in entomological research. The integration of field and laboratory observations, along with quantitative assessments of porosity, damage rates, and developmental timing, provides a solid methodological foundation. The tables and figures are generally clear, well-organized, and supportive of the results. In particular, the correlation analyses offer important insights into morphological constraints during development. Finally, the study provides practical recommendations for integrated pest management, such as optimized pruning schedules, which are of direct relevance to viticultural practice.

Suggestions for Improvement:

Language: The manuscript contains several grammatical and typographical errors, as well as some overly long or redundant sentences. A thorough proofreading and language edit is recommended to improve clarity and readability.

Title: Consider shortening the title for conciseness. A possible revision could be:
“Developmental Biology and Seasonal Damage of the Grape Borer Xylotrechus pyrrhoderus in Grapevines.”

Abstract: The abstract should better reflect the study’s methodology—particularly the innovative use of imaging tools—and highlight the practical applications of the findings more explicitly.

Methodological Clarity:

Clearly specify the sample sizes used in statistical analyses (e.g., developmental timing).

Discussion: Strengthen the discussion by connecting the results to prior studies on similar wood-boring pests and the use of imaging in entomology.

The finding that adults remain inside branches for approximately 14 days after emergence is particularly interesting. The authors should explore possible adaptive explanations for this behavior, such as its role in sexual maturation, predator avoidance, or environmental buffering.

Consider adding a paragraph comparing the life cycle of X. pyrrhoderus with that of related cerambycid beetles, which would help contextualize the findings within a broader ecological and taxonomic framework.

Author Response

Response to Reviewer 1:

Comments 1: The manuscript contains several grammatical and typographical errors, as well as some overly long or redundant sentences. A thorough proofreading and language edit is recommended to improve clarity and readability.

Response 1: We have proofread and improved the language in the paper. These changes can be viewed throughout the paper.

Comments 2: Title: Consider shortening the title for conciseness. A possible revision could be: “Developmental Biology and Seasonal Damage of the Grape Borer Xylotrechus pyrrhoderus in Grapevines.

Response 2: We have changed the title to: Developmental Biology and Seasonal Damage of the Grape Borer Xylotrechus pyrrhoderus in Grapevines in line 2-4 as suggested.

Comments 3: Abstract: The abstract should better reflect the study’s methodology—particularly the innovative use of imaging tools—and highlight the practical applications of the findings more explicitly.

Response 3: The abstract was modified to highlight more the new methodologies used in the manuscript and their relevance for future practical applications in line29-51.

Comments 4: Methodological Clarity: Clearly specify the sample sizes used in statistical analyses (e.g., developmental timing).

Response4: We have added “In this experiment, grape borers in 160 branches were tracked and observed.” in line 159.

Comments 5: Discussion: Strengthen the discussion by connecting the results to prior studies on similar wood-boring pests and the use of imaging in entomology.

Response 5: We have modified the discussion and further cited previous research and compared it to our results, as suggested by the reviewer. Please refer to lines 318-333.

Comments 6: The finding that adults remain inside branches for approximately 14 days after emergence is particularly interesting. The authors should explore possible adaptive explanations for this behavior, such as its role in sexual maturation, predator avoidance, or environmental buffering.

Response 6: Thank you for this comment. We have further discussed this point and compared it to previous research and added references in line 370-371、382-384.

Comments 7: Consider adding a paragraph comparing the life cycle of X. pyrrhoderus with that of related cerambycid beetles, which would help contextualize the findings within a broader ecological and taxonomic framework.

Response 7: We have added a paragraph as suggested. Thank you for the comment. Please refer to lline 344-366.

Reviewer 2 Report

Comments and Suggestions for Authors

Dear Authors,

I would like to express my appreciation for submitting this manuscript.
The topic, the application of modern techniques, and the results obtained are commendable.

However, the manuscript, in its current form, requires significant improvements in presentation and clarity.

Several technical issues, such as misspellings and formatting inconsistencies, should be easily corrected. These have been marked in the manuscript PDF.

Additionally, some appendices, clarifications of the text, and restructuring of certain sections could enhance the clarity and readability of your results. All remarks are presented in PDF.

I strongly recommend implementing the indicated corrections and revisions before the manuscript can be considered for publication.

Best regards,

Author Response

Response to Reviewer 2:

Comments 1: enhance our

Response 1: We have deleted “enhanceour”, and replace with “enhance our” in line 23.

Comments 2: Will you reconsider use of appropriate terms, please.

Response 2: We have deleted “branches”, and replace with “canes” in line29. Furthermore, these changes can be viewed throughout the paper.

Comments 3: Is it correlated to shoot diameter?

Response 3: We also suspect that it is correlated to the diameter of the branches. We are currently conducting research on this issue, including aspects such as variety, nutrition, and branch diameter.

Comments 4: Provide information of reduction of vine production, % please. Also, you should quoted vineyard surface in those three countries, and part in world wine production according to availabe FAOStat data. In this way will be underline economic significance of this pest.]

Response 4: We have added “According to the data of the Food and Agriculture Organization, China is a major country in grape cultivation and wine production, with a vast planting area and considerable wine production; the grape planting areas in Japan and Korea are relatively small, and their wine industries are of limited scale” in line 61-64; We added “When the damage is serious, the infestation rate of grape plants can reach over 85%.” in line 66-67.

Comments 5: Can you provide some actual publish data which could be support written sentence. Quoted reference are dates back in 90 years of XX century, and it could be considered only as  out of date or non supportive to presented facts.]

Response 5: We have supplemented some references in line 67.

Comments 6: IS it present or introduce worldwide? In that sense significance of research will be global not just oriented to East Asia.]

Response 6: Currently, there are records in the literature only in China, Japan, and Korea. We have supplemented some references in line 71.

Comments 7: 1 year old branches. Be precise, is the shoot or cane

Response 7: We agree with this comment. We have deleted “1-year-old grapevine branches”, and replaced with “grape canes” in line 72.

Comments 8: During which month?

Response 8: We have added “during August to September” in line 73-74.

Comments 9: When this insect activities are fulfilled in time frame, i.e. month.

Response 9: There has been very little research on this insect, so much of its biology remains unclear. We are currently conducting research on its mating and egg - laying habits.

Comments 10: Do you think copulation as a term?

Response 10: We agree with this comment. We have made modifications in line 74-75.

Comments 11: WHEN IS OVIPOSITION? Biology of the species is unclear.

Response 11: There has been very little research on this insect, so much of its biology remains unclear. We have deleted Following copulation,”, and replaced with “On the day of mating, the female can lay eggs.” in line 74.

Comments 12: You should represent pest biology clear and briefly.  This is embryogenesis, BUT reconsider sentence reconstruction. So, first female oviposits eggs, how , once or oviposition registered twice or three times. Were eggs on vine, as cluster or single. After oviposition eggs are under process of embryogenesis 5-7 days. Larvae hatch during which month, punctate and  boring the shoot and continue development inside the shoot.]

Response 12: There has been very little research on this insect, so much of its biology remains unclear. We have added “Each female lays one egg at a time.” in line 76.

Comments 13: Precise larval stage.

Response 13: Much of the biology of this insect is unstudied, including the larval stage, so we cannot pinpoint the specific larval age. We are currently researching the age of the insect, but because the small (young) larvae of the insect can only survive in the wild branches and cannot be raised indoors, the study is still difficult and will take some time to complete.

Comments 14: Decay or dieback

Response 14: We have deleted “death”, and replaced it with “dieback” in line 81.

Comments 15: It can not be NEW shoots. Larvae feeding inside the same shoot in which overwinter. So, some colonized shoot wilting and decay in May of next year. Some larvae continue development until July, could be concluded. It is interesting to underline in which type of shoots larvae are fully develop, thiner or thicker.

Response 15: It may be that our expression is unclear, so we redescribe it. We have deleted “the wilting or death of new shoots”, and replaced it with “the grape buds do not germinate in spring, or winer after germination, and in severe cases, a large number of branches and vines dieback” in line 80-82.

Comments 16: Stipulate duration of this stage please.

Response 16: We have deleted “in July”, and replaced with “from early July to the end of August.” in line 83-84.

Comments 17: how many days before eclosion imagoes spend in the shoots? Are imago mature for copulation? How many days they are fly and copulate? When female oviposit eggs.

Response 17: The time of adult insects inside the shoots, which we studied later in this article, is about 14 days in Table 1 in lin263 We think they are mature because they can mate and lay eggs the same day they leave the shoots. This is proved by our studies on the development of adult ovaries at different stages of staying and leaving the branches, which will be presented in another article.

Comments 18: ,, reon

Response 18: We have deleted “,, reon”, and replaced it with “,research on” in line 90.

Comments 19: [..]

Response 19: We have deleted “.”” in line 94.

Comments 20: Support this data with some reference, please.

Response20: We have added the reference in line 98.

Comments 21: branches and plants

Response 21: We have deleted “branches”, and replace with “canes” in line131.

Comments 22: larvae grow and feed.

Response 22: We rephrased this sentence and now it is: "as larvae grow and start feeding" in line 181-182.

Comments 23: symptom and it can be used as indicator

Response 23: We have deleted “indicator”, and replace with “symptom” in line 183.

Comments 24: wilting or death

Response 24: We have deleted “death”, and replace with “dieback” in line 184.

Comments 25: It is repeated n next sentence

Response 25: We have deleted “, causing branch wilting or death.” in line 186.

Comments 26: conditionsshowed Ad blank/space please.

Response 26: We have added space in line 193.

Comments 27: after the damage

Response 27: We have deleted “after the damage” in line 194.

Comments 28: This sentence is not clear. Or question is what you intend to comment.

Delete, please.

Response 28: We have deleted “alone resulted in a noticeable reduction in damage.” in line 196-197.

Comments 29: increase significantly greater

Response 29: We have deleted “significantly greater” in line 198.

Comments 30: It is probably shown in Figure 2B. You also should explain this data as some consequence, insect biology or behaviour.

Response 30: We have deleted “The”, and replaced it with “In Figure 2B, the assessment of grapevine damage under different treatment conditions showed that the damage rate of branches increased sharply without any treatment (control) of X. pyrrhoderus. Specifically, as X. pyrrhoderus developed and continued to feed, this upward trend in the damage rate became even more pronounced.” in line 192-197.

Comments 31: Italic, please.

Response 31: We have italicized in line 199 as suggested.

Comments 32: Authorise photo please. It is original, I suppose.

Response 32: We did not fully understand what the reviewer means by authorizing the photo. The photo is indeed original.

Comments 33: space or blank

Response 33: We have added space in line 211.

Comments 34: delete dot

Response 34: We have deleted dot in line 220.

Comments 35: Italic

Response 35: We have made this modification in line 230-233.

Comments 36: Delete extra comma.

Response 36: We have deleted extra comma in line 244.

Comments 37: space or blank

Response 37: We have added space in line 246.

Comments 38: This is completely misunderstanding of development. Larva start development in last year, maybe this is correct to compare development from the larvae transfer to new shoot, but it can not be analyzed in this way. You have to analyzed duration of development only of last larval instar (stage).

Response 38: We did not analyze the time of larval development in this study, and the specific reasons have been explained in the discussion section in line 387-392. In addition, as replied in comment 13, we have not yet clarified the age of the insect, so it is difficult to judge the time of the last instar.

Comments 39: Chamber is probably highly depend of shoot diameter, but you did not measure it. In that sense, larval size is actually dependent of shoot diameter. Also, vigorous shoots are probably severe infested, but this data missing also.

Response 39: We also suspect that it is correlated to the diameter of the branches. We are currently conducting research on this part, including aspects such as variety, nutrition, and branch diameter.

Comments 40: it is trunk for the vine or grape plant.

Response 40: It is grape canes. We have added “of grape canes” in line 297-298.

Comments 41: Delete extra dot, please.

Response 41: We have deleted extra dot in line 301.

Comments 42: Appropriate term, please. withered branches

Response 42: We have the change the word "withered" with "wilting", to us it might be more appropriate in line 303.

Comments 43: How? Are larvae leave withered shoots and colonise new i.e. healthy shoots in the spring? You have to be completely clear in comments and conclusion, because this represent crucial data for practical implementation of spring pruning, and lead to decrease of pest population and damage in vineyard.reduce larvae - reduce larvae population is maybe better expression.

Response 43: We have added “The larvae are weak in mobility and do not leave the original canes. The following spring, it will continue to feed on the original canees and cause damage. The purpose of spring pruning in the following year is that the damage caused by larvae in the previous winter is not obvious. Some damaged parts cannot be detected during the winter pruning. However, in the spring of the following year, the damage caused by larvae becomes distinct, and thus canes infested with insects can be further pruned and removed”in line 303-310.

Comments 44: Split this long sentence, please.

Response 44: We have deleted “suggesting that our study addresses this concern of effects on larval development.” in line 337-338.

Comments 45: The main question the grape grower asks is: WHEN in the growing season.

The time or BBCH of vine missing for this sentence.

Response 45: Thank you for pointing this out. We have added “Considering the grapevine's growth cycle, this occurs during the period when the grapes are starting to change color” in line in 369-370.

Comments 46: Based on what do you conclude this.

Response 46: Thank you for pointing this out. This phenomenon is what we observe. We put the male and female adults that have just left the branches into the breeding box, and at the same time put in fresh grape branches, and we can see the male and female mating and laying eggs on the branches on the same day. This part will be described in detail future articles dealing with oviposition behavior of X. pyrrhoderus. We have added “in the study we found” in line 379.

Comments 47: space or blank

Response 47: We have added space in line 386.

Comments 48: What is this? The whole sentence should be reconsider.

Response 48: We have made modifications in line 388.

Comments 49: Sentence not clear. split stages when

Response 49: We have added space in line 395.

Comments 50: delete extra comma

Response 50: We have deleted extra comma in line 398.

Comments 51: delete

Response 51: We have deleted “improving” in line 402.

Comments 52: This is completely expected, because last larvae instar/stage finishing development and their need for food increase. BUT, here you have recommend additional measure to mitigate vineyard looses.

Response 52: While it is true that the increase in larvae's food demand with their development contributes to the higher damage rate in May, it's important to note that May is not the final instar stage of this pest. Considering the continuous development and potential harm of the larvae throughout different stages, implementing additional pest management strategies in spring and summer, aside from traditional winter pruning, is crucial. These supplementary measures are designed to proactively mitigate potential vineyard losses at various development phases of the pest, ensuring more comprehensive protection of the vineyards.

Comments 53: You have to finish this idea.  In area of investigation it is WHEN or quote BBCH stage of vine.

Response 53: Thank you for pointing this out. We have added “during the period when the grapes are enlarging and starting to change color” in line 419.

Round 2

Reviewer 2 Report

Comments and Suggestions for Authors

Dear Authors,

Thank you for your prompt, accurate, and comprehensive corrections.

A minor technical issue has been identified in the text and marked for correction.

Best regards,

Author Response

Dear Editor,

The second reviewer suggested a few minor corrections, which were included in the revised manuscript but did not appear in the track-changes version. These corrections have now been incorporated into the final version.

Thank you.